Modeling LTE-advanced cell capacity estimation using packet bundling and carrier aggregation

Senapati Rajiv rajiv.s@srmap.edu.in
Department of CSE, SRM University, Andhra Pradesh , Amaravati, Andhra Pradesh , India
Sadek Rowayda
Electronic publication date: 2025 May 23
Publication date: 2025
Volume: 11
Electronic Location ID: e2868
Received 2024 Sep 17; Accepted 2025 Apr 10
Copyright: © 2025 Senapati
Copyright year: 2025
Copyright holder: Senapati
License: This is an open access article distributed under the terms of the Creative Commons Attribution License, which permits unrestricted use, distribution, reproduction and adaptation in any medium and for any purpose provided that it is properly attributed. For attribution, the original author(s), title, publication source (PeerJ Computer Science) and either DOI or URL of the article must be cited.
License URL: https://creativecommons.org/licenses/by/4.0/

Keywords: Carrier aggregation, Cell capacity, CQI, LTE-advanced, Packet bundling

Funding: The authors received no funding for this work.

==============================
The increasing demand for mobile usage raises many challenges for service providers. Satisfying subscribers by providing a better quality of service (QoS) is one of the major concerns of an operator. One way to solve this problem is to adopt an efficient radio resource use mechanism. In this context, smart network planning management requires the estimation and enhancement of capacity in terms of the number of subscribers within a cell for better QoS provisioning through radio resource usage optimization. The model proposed in this article explicitly uses some long term evolution (LTE)-Advanced (LTE-A)-specific enhancements such as carrier aggregation (CA) and channel quality indication (CQI)-based resource allocation. Further, this study proposes a CQI-based clustering approach with packet bundling and CA to optimize radio resource utilization and enhance LTE-A cell capacity. LTE-A cell is logically divided into clusters such as the Silver class, Platinum class, Gold class, and Diamond class based on the CQI from the user end to eNodeB (eNB). Further, cell capacity (CCa) estimation algorithms are proposed in a simplistic scenario as well as in each cluster using packet bundling factor ω considering CA. From the result analysis, it is found that appropriate modulation and voice codecs can be used in appropriate clusters to enhance the cell capacity. Furthermore, it is observed that the packet bundling factor helps in improving the radio resource usage and thereby improving the capacity of a cell. The research work proposed in this article can be extended further to estimate the user capacity in the context of the 5th generation cellular network.

Introduction

Long-term evolution-Advanced (LTE-A) extends most of the capabilities of its predecessor 3GPP (2009). It is expected to provide higher capacity in terms of data rate, spectral efficiency, and high cell capacity (CCa) in terms of subscribers, as well as improved performance at cell edge (Wännström, 2013). Carrier aggregation (CA) is one of the key technologies adopted in LTE-A network for bandwidth expansion by the use of the multi-carrier technique to support the deployment bandwidth up to 100 MHz (Liao, Chen & Chen, 2014; Zhang et al., 2014; Radhakrishnan, Neduncheliyan & Thyagharajan, 2019; AlQahtani, 2017). In literature, CA has been extensively addressed from different preservatives as a mean of bandwidth expansion in LTE-A network (Zhang et al., 2011; Zheng et al., 2014; AbdelFadeel et al., 2016; Rostami, Arshad & Rapajic, 2017; Ratasuk, Tolli & Ghosh, 2010; Kamath, Singh & Khanna, 2020). It is one of the straightforward ways to improve the CCa of a network by expanding the bandwidth. In this technique, each aggregated carrier is referred to as component carrier (CC), that operates in various system bandwidths of long term evolution (LTE) (i.e., from 1.4 to 20 MHz) and a maximum of five CC can be aggregated to extend the total bandwidth up to 100 MHz.

CA is one of the promising technologies for 4th generation (4G), 5th generation (5G), and beyond networks. In general, CA combines more than one carrier into a single data channel to improve the data capacity. CA creates new opportunities to make better use of the spectrum using frequency bands together to deliver more capacity. CA was introduced in Third Generation Partnership Project (3GPP) Rel. 10 in 2011 and incorporated in LTE-A. By aggregating multiple channels together the operators bandwidth (BW) may increase and in turn increase the capacity of the network.

In this article, the LTE-A cell capacity is modeled using packet bundling and CA. Packet bundling and CA are both techniques used to improve the data transmission efficiency in LTE and LTE-A, but they serve different purposes and operate at different levels of the communication stack. CA is a radio-level technique used in LTE-A to increase the bandwidth and improve data rates by combining multiple frequency carriers. LTE has a maximum bandwidth of 20 MHz per carrier; however, LTE-A supports up to five aggregated carriers, reaching a total bandwidth of 100 MHz. Packet bundling is a MAC-layer technique that groups multiple small packets into a single transmission unit to reduce overhead and improve efficiency, especially in low-latency applications.

Motivation

Today, staying connected is more important. Speed and good user experience are equally important. We are living in the invention age, and the adequate technology is also limited. We are witnessing massive transformation and development in the industrial sector, smart education system, healthcare, automotive industry, smart economy, smart lifestyle, and many more. In particular, the exponential growth of mobile users raises many challenges for the operators in terms of quality of service (QoS). To retain the desired QoS, operators must satisfy the subscriber’s requirement by providing adequate radio resources. Physical radio resources are one of the precious entities in wireless communication, and it is essential for the network operator to optimize the radio resource usage properly. This essentially requires a mechanism to estimate and enhance the active subscribers within an LTE-A cell. Hence, in this article, CCa estimation model and algorithm is proposed in a simplistic scenario as well as using the packet bundling approach considering CA.

4G-LTE Advanced is still of significant importance for several reasons. Although 5G is the current focus, 4G continues to play a crucial role in the global cellular communication landscape. As per the work reported in Saha & Cioffi (2024), dynamic spectrum sharing between LTE and 5G New Radio has become an important aspect where both systems can co-exist on the LTE spectrum. The work reported in Wang & Xu (2023), study has been conducted on shared aperture 4G LTE and 5G mm-wave antenna in mobile phones with enhanced mm-wave radiation in the display direction. In Zhao & Wang (2024), a mobile antenna with a 0.5-mm clearance, which operates in 11 bands, is proposed for 4G and 5G metal-bezel smartphones. From the literature, it is found that 4G LTE is still an area of interest, and especially in rural and under-served areas, it is the most cost-effective. However, research is continuing to improve the performance of 4G in terms of efficiency, coverage, and QoS specifically for those under-served regions.

Contributions

The major contributions of this article are outlined as follows: The proposed model explicitly uses some LTE-A-specific enhancements i.e., beyond standard LTE features such as CA and channel quality indication (CQI)-based resource allocation.

The study proposes a CQI-based clustering approach with packet bundling and CA to optimize radio resource utilization and enhance LTE-Advanced cell capacity.

To be able to optimize the radio resource utilization, it is proposed to divide the LTE-A cell logically into clusters such as the Silver, Platinum, Gold, and Diamond classes. These clusters are formed based on the reported CQI by the user equipment (UE) to eNodeB (eNB).

An LTE-A CCa estimation model for voice traffic using CA is developed in this article considering a simplistic scenario and then LTE-A CCa estimation algorithm is proposed based on that model.

Further, LTE-A CCa estimation model is proposed using Packet bundling. Using the proposed model, LTE-A modulation selection and capacity estimation algorithm is proposed.

Finally, the impact of different factors on CCa is discussed through the result analysis.

Related works

CA has been well studied in the literature addressing the 4G/5G network for CCa enhancement in terms of throughput. However, CCa estimation in terms of the number of voice subscribers using CA is not well explored. Different metrics have been considered as an indicator of CCa. In this article, LTE-A CCa is defined as the maximum limit of voice mobile users that can be served at any particular instance within a single LTE-A cell. However, in earlier literature (Arteaga et al., 2014; Carpin et al., 2015; Gomez et al., 2015; Hidayat et al., 2017; Jha et al., 2017; Mahfoudi et al., 2014; Østerbø, 2011; Jyothi & Chaudhari, 2022), throughput based CCa is studied, which is a physical (PHY) layer estimate. Such estimation helps to determine the data carrying capability of the channel, which is largely dependent on the modulation scheme used. However, these estimates do not consider the protocol overheads, delay requirements, and packet losses. Therefore, it may not always be a suitable option to design call admission control (CAC) based on the PHY layer estimate (i.e., the capacity of the channel in terms of throughput or bit rate). In LTE like networks, the basic resource elements are the PRBs, which are designed to be allocated and managed at the MAC layer. This leads to make a physical resource block (PRB) based estimate of the CCa. Some earlier schemes have also estimated the capacity in terms of concurrently served users for voice over long term evolution (VoLTE) service (Wang, Jiang & Tuomaala, 2007; Senapati & Pati, 2018, 2019a, 2019b; Senapati, 2021; Fan & Valkama, 2012; Song et al., 2005; Jouihri & Guennoun, 2016; Corbun, Almgren & Svanbro, 1998; Bessette et al., 2002; ElNashar, 2014). In Shin et al. (2023), the evolution of V2X from 4G to 5G in 3GPP is reviewed, focusing on the resource allocation aspects defined in LTE and new radio (NR) specifications. The work reported in Baharin et al. (2024), analyzed the impact of microwave radiation pulses emitted from natural lightning on 4G LTE mobile communication data transmission. Further such research could be useful for improving the reliability and performance of 4G LTE networks, particularly in areas prone to thunderstorms. In Hajjar et al. (2017), a new algorithm for relay selection in a multi-cell scenario based on K-means and a selection strategy is proposed for capacity enhancement in the LTE cell.

A simple PRB based analytical model is proposed in Wang, Jiang & Tuomaala (2007) for estimating the voice user capacity in uplink (UL) using adaptive multi-rate (AMR) voice codec. In Senapati & Pati (2018), eNB capacity in duplex mode is calculated by a simplistic scenario using adaptive multi-rate wideband (AMR-WB) voice codec. The result shows that lower order voice codec gives higher capacity as compared to higher order voice codec. Theoretical capacity limit from the perspective of radio resource utilization is estimated and found reaching its upper limit in Fan & Valkama (2012). The work reported in Jouihri & Guennoun (2016) shows the bottleneck of voice over internet protocol (VoIP) capacity. From the simulation results obtained through the open wireless network simulator (OWNS) concludes that the worst method provides the highest VoIP capacity. The work reported in Corbun, Almgren & Svanbro (1998) describes the capacity and speech quality aspects using AMR codec. It is found that there is a degradation in speech quality with increasing capacity. However, a potential gain in speech quality is found in AMR codec as compared to the existing codecs in global system for mobile communications (GSM). The work reported in Bessette et al. (2002) suggests AMR-WB codec for voice communication in LTE network due to its extended audio bandwidth from 50 Hz to 7 kHz, which gives better speech quality compared to AMR-NB codec. In ElNashar (2014), VoIP CCa is estimated using AMR-NB voice codec. This estimation assumes a specific modulation and coding for every VoIP packet in an error-free scenario. However, the higher-order modulation scheme was not considered in this study. In Bae et al. (2009), number of PRBs required for a voice service request while using a specific modulation and coding scheme (MCS) is evaluated for designing resource estimated dynamic CAC algorithm. Since the algorithm considers the minimum data rate requirement, it can maximize the use of radio resources. In Salman et al. (2017), several spectrum sensing approaches for LTE and LTE-A systems are investigated in the cognitive radio system for improving the user with high speed data rate. In Ali & Nazir (2018), a comprehensive review of radio resource management techniques, scheduling, and QoS is carried out to optimize resource allocation.

A new QoS and channel-aware packet bundling mechanism for improving spectral efficiency using adaptive modulation and coding is presented in Choi, Kim & Beard (2010). Through this study, they observed the improved channel usage while using packet bundling in similar channel conditions. The work reported in Torres Compta, Fitzek & Lucani (2015) presents the strategies to manage the heterogeneous kinds of packet length by appropriate packetization using packet bundling, which can further reduce the overhead associated with the data packet. In Kim et al. (2017), data aggregation and packet bundling are used in UL small packets for monitoring LTE applications. Furthermore, from the result, they found that packet bundling alleviates the overhead and optimizes the resource usage while sending small packets in UL. From the literature, it is observed that packet bundling is used for improving the spectral efficiency and throughput. However, optimizing the accommodation of active subscribers within a cell using packet bundling requires further investigation.

The articles found in the literature have studied the CCa by applying some of the modulation and codec techniques, and in some cases, they have tried with some of the older codec techniques. However, finding LTE-A CCa in terms of voice users considering CA needs to be investigated. Further, the CCa changes from cluster to cluster due to varying signal quality. This makes the capacity estimation process a challenging task. To realize this goal, a mathematical model is proposed to estimate the capacity in terms of active users in an LTE-A network. Further, an algorithm for LTE-A capacity estimation using CA is proposed based on the analytical model. In this model, CQI index is used to classify the cell regions into the Silver class cluster, Platinum class cluster, Gold class cluster, and Diamond class cluster corresponding to excellent, good, average, and poor radio conditions. This classification helps in assigning appropriate adaptive modulation and coding (AMC) and codec techniques in different clusters for efficient radio resource utilization as well as for capacity enhancement.

The use of a large number of transmit and receive antennas can certainly achieve better spectral efficiency through multiple input multiple output (MIMO) and massive MIMO (Vasudevan et al., 2023). However, in certain areas with geographical complexities, deploying MIMO requires significant investment in hardware and infrastructure. In such a situation, the proposed approach may be useful to accommodate the maximum number of users with a given system bandwidth.

Organization of article

The rest of this article is presented as follows. The system model is presented in “Related Works”. LTE-A CCa estimation model considering CA is presented in “System Model” followed by an algorithm. Resource distribution criteria and model parameters are presented in “LTE-A Capacity Estimation Model and Algorithm using Carrier Aggregation”. Numerical results are provided with a detailed discussion in “Resource Distribution Criteria and Model Parameter”. Finally, the conclusions are presented in “Numerical Results”.

System model

A single LTE-A cell (considered as a primary cell) is taken into consideration in this work. The primary cell users are grouped into different clusters i.e., Silver class, Platinum class, Gold class, and Diamond class based on their radio condition. The radio condition of the user is obtained using the CQI index reported by the user equipment (UE) to the serving eNB. Then appropriate MCS may be assigned to the subscriber based on their CQI.

CQI is primarily influenced by signal-to-noise-plus-interference ratio (SINR), Block Error Rate (BLER), Doppler Frequency Shift, and Path Loss (PL). Mathematically CQI can be expressed as:

(1) CQI=⌊a.log2(1+b.SINR)+c.(1−BLER)−d.fd⌋

where, SINR is the received signal power over interference and noise, which can be expressed as: SINR=PsI+N, where Ps is the received signal power, I is the interference power, and N is the noise power, block error rate (BLER) is measured through hybrid automatic repeat request (HARQ) feedback and represents packet error probability. LTE-A defines CQI such that the chosen MCS maintains a BLER≤10%. The CQI value is mapped to the highest MCS that satisfies this condition. Doppler frequency shift, the mobility factor fd can be expressed as fd=vfcc, where v is UE velocity, fc is the carrier frequency, and c is the speed of light. A higher fd can degrade CQI. a,b,c,d are the tuning parameters based on empirical network settings. The CQI values range from 1 to 15, with higher values indicating better channel conditions. Based on CQI values, the LTE-A cell is divided into Silver (CQI 1-3), Platinum (CQI 4-6), Gold (CQI 7-11), and Diamond (CQI 12-15) clusters, ensuring optimized resource allocation based on real-time channel conditions. Here MCS can be considered as a function of CQI as it is reported in 3GPP TS 36.213 specification (3GPP, 2011). The function can be expressed as:

(2) f(CQI)={QPSK1≤CQI≤3164QAM4≤CQI≤664QAM7≤CQI≤11256QAM12≤CQI≤15.

The research work is based on the idea that user equipment that reports CQI between 1 and 3 belongs to a group called the Silver class. Voice transmission uses lower order modulation like quadrature phase shift keying (QPSK) within this group. Similarly, the equipment that is reporting CQI within the range of 4 to 6 forms a cluster called Platinum class, and 16 quadrature amplitude modulation (16QAM) modulation is used within that cluster for voice transmission. Further, the equipment that is generating CQI within the range of 7 to 11 is grouped in a cluster called Gold class, and 64 quadrature amplitude modulation (64QAM) modulation can be used within the cluster for voice transmission. Finally, the equipment that is generating CQI within the range of 12 to 15 is grouped in a cluster called Diamond class, and 256 quadrature amplitude modulation (256QAM) can be used in that cluster for voice transmission. In this study, a single base station is considered in a rural setup with no interference with the objective of accommodating a larger number of users with limited or existing network resources while using 4G LTE services.

Radio conditions within a cluster are assumed to be the same, and within that cluster, all the users are assigned the same modulation and coding for voice packet transmission. Using the above scenario, the number of voice subscribers that can be accommodated within an LTE-A cell is estimated in this work. Figure 1 represents a primary cell, and a UE is listening to that cell on a carrier. This carrier is from 1.4 to 20 MHz. In case CA is enabled, there will be another cell, which is known as a secondary cell. Now UE will listen to the secondary carrier also. In this case, a UE can listen to 2 CCs. So let us say that the first CC is 20 MHz and the 2nd CC is 20 MHz, then the total BW within UE and eNB is 40 MHz while using CA. In this way, there can be a maximum of 5 CCs that an UE can support. If a UE will support a maximum of 5 CCs, then in that case one CC will be the primary component carrier (PCC) and the other 4 CCs will be secondary component carrier (SCC). In that case, the eNB will support five cells, and the maximum bandwidth a UE can avail is 100 MHz. It is obvious that with maximum bandwidth, the throughput will be maximum. However, the capacity in terms of the number of UEs is calculated for this work. It is quite obvious that the signal conditions throughout the cell may not be the same. The UE close to the eNB experience good signal, and the UE far from the eNB may experience bad signal quality. In such a case, it is very important to categorize the UE based on the position of the UEs and then to allocate the appropriate modulation and coding scheme. The idea behind using a large quantization size of four levels in Fig. 1 is to explore if a finer granularity could provide any addition in terms of capacity.

Figure 1 Clusters based on the CQI.

Lte-a capacity estimation model and algorithm using carrier aggregation

In this section, the proposed capacity estimation algorithm is presented for voice service under the LTE-A cellular network using CA. The analytical models are developed considering the system model, which is explained in “Related Works”. Using the proposed analytical model, the LTE-A CCa estimation algorithm at downlink for voice service using CA is proposed first using a simplistic approach, and then the capacity estimation model is presented using a packet bundling-based approach followed by an algorithm.

LTE-A CCa estimation model at DL for voice service using CA

In this section, an analytical model is presented to estimate the number of subscribers that can be accommodated within a cell using CA. In this model, the CCa is estimated by dividing the total number of physical resources available in a given bandwidth and the resources required for a subscriber to establish a connection and transmit a voice packet. The capacity estimation model is calculated using the following expressions. All the symbols used in this model are presented in Table 1.

Table 1 Symbols and acronyms with description.

Symbol	Description	
NBW_cc	System bandwidth	
NRB_ccsc	SC within a resource block	
BWccsc	BW of a sub-carrier	
NRB_cc	PRB for a specific BW	
RBtotal_cc	PRB available for voice transmission in a given BW	
RBccreqd	PRB needed to carry voice payload	
NRB_ccTTI	Number of physical resources per TTI	
NTTI_ccVoLTE	TTIs per 20 ms time frame	
Ravr_re	Avg. re-transmission	
Pv	Payload of voice packet	
Cr	Bit rate of voice codec used	
Vt	LTE-A frame time length for voice packet	
Ov	Voice packet overhead	
Oh	RObust Header compression	
PVoLTE	Payload for LTE-A voice packet	
PMCS	Payload of modulation scheme used	
n	No. of CC to be aggregated	
RBused	No. of used RBs in UL/DL	
f	Resource distribution factor	
Nave	Physical resources required for voice packet	
Nave′	Physical resources required for SID packet	
1−p	Proportion of voice packet arrival in active state	
p	Proportion of voice packet arrival in a nonactive state	
v	Voice activity factor	
ω	Bundling factor for voice packet	
ω′	Bundling factor for SID packet	
NSupcc	Total number of supported users in a single CC	
NSup	Total number of supported users considering n CC.	
ωQPSK	Packet bundling factor while using QPSK	
ω16QAM	Packet bundling factor while using 16QAM	
ω64QAM	Packet bundling factor while using 64QAM	
ω256QAM	Packet bundling factor while using 256QAM	
UEMCS	Modulation scheme assigned to the user	

The total number of resource blocks for a slot NRB_cc in a particular bandwidth is a fraction of the given bandwidth NBW_cc and the product of subcarriers available in a resource block and the bandwidth of a subcarrier, which is expressed in Eq. (3).

(3) NRB_cc=NBW_ccNRB_ccsc×BWccsc.

The maximum number of available resource blocks that can be used for packet transmission can be calculated using Eq. (4).

(4) RBtotal_cc=NTTI_ccVoLTE×NRB_ccTTI.

The voice payload is calculated using Eq. (5).

(5) Pv=Cr×Vt+Ov.

The target payload i.e., the combination of voice payload and the overhead can be calculated using Eq. (6).

(6) PVoLTE=Pv+Oh.

Finally, the resource block required to carry the target payload can be calculated using Eq. (7).

(7) RBccreqd=⌈PVoLTEPMCS⌉.

Now, LTE-A CCa in downlink (DL) denoted by Nsupcc for a CC can be expressed as given in the following expressions.

(8) Nsupcc≤{∑i=14RBtotal_ccRBcc,ireqd,if∀iRiavr_re=0∑i=14RBtotal_ccRBcc,ireqd×(1+Riavr_re),otherwise

where RBcc,ireqd denotes the physical resources required by each voice payload in ith cluster and Riave_re denotes the average re-transmissions required in ith cluster.

Now, the capacity of an LTE-A cell considering CA can be defined as:

(9) Nsup=n×Nsupcc.

where n denotes the number of carrier components needs to be aggregated by an eNB in an LTE-A network.

Proposed algorithm

The proposed algorithm presented in Algorithm 1 facilitates finding the number of voice subscribers accommodated within a cell in DL. Further, CA is used to enhance the accommodation of voice users within the cell. The algorithm is developed considering realistic radio conditions within a cell. It means the radio condition varies within a cell, and accordingly, the subscriber requires radio resources to establish a connection. To simplify the resource allocation process, a channel quality indicator is used to classify cells into cluster-1, 2, 3, and 4. Based on the channel quality within each cluster, appropriate modulation is used, and the physical resource requirements can also be calculated.

Algorithm 1 LTE-A cell capacity estimation using carrier aggregation.

1: NRB_ccTTI=2×NBW_ccNRB_ccsc×BWccsc.	
2: RBtotal_cc=NTTI_ccVoLTE×NRB_ccTTI.	
3: Pv=Cr×Vt+Ov.	
4: PVoLTE=Pv+Oh.	
5: Nsupcc = 0.	
6: For (i = 1; i ≤ 4; i++)	
7: if (i == 1) then /* Diamond class Cluster */	
8: UEMCS← 256QAM.	
9: RBcc,ireqd=⌈PVOLTEPMCS⌉.	
10: end if	
11: if (i == 2) then /* Gold class Cluster */	
12: UEMCS← 64QAM.	
13: RBcc,ireqd=⌈PVOLTEPMCS⌉.	
14: end if	
15: if (i == 3) then /* Platinum class Cluster */	
16: UEMCS← 16QAM.	
17: RBcc,ireqd=⌈PVOLTEPMCS⌉.	
18: end if	
19: if (i == 4) then /* Silver class Cluster */	
20: UEMCS← QPSK.	
21: RBcc,ireqd=⌈PVOLTEPMCS⌉.	
22: end if	
23: Nsupcc+=f×RBtotal_ccRBcc,ireqd×(1+Riavr_re). /* using Eq. (8) */	
24: end For	
25: Nsup=n×Nsupcc⌋. /* using Eq. (9) */	

The step-by-step execution of the proposed algorithm is presented in the following. First, the total available radio resources for voice packet transmission within a given bandwidth are calculated, followed by determining the payload size using the AMR-WB encoder and incorporating RObust Header compression (ROHC). The resource requirements for voice packets are then computed separately for each cluster, allowing the estimation of the number of voice calls supported within the LTE-A cell in DL. Finally, considering CA, the total number of accommodated voice calls is determined based on the number of CC aggregated.

In this proposed algorithm, a fixed percentage of resources are allocated to each cluster. However, a dynamic resource allocation mechanism can be used in real time based on traffic demand, channel condition, and user distribution. The same can be implemented by continuously monitoring the number of UEs in each cluster and adjusting resource allocation proportionally to the active UEs in each cluster. Furthermore, we may adopt a threshold-based approach, in which clusters with higher demand receive additional resources from underutilized clusters. Mathematically, the model can be expressed as Ri=Ni∑j=14Nj×R, where Ri is the allocated resources for cluster i. This procedure ensures fair allocation based on real-time demand. A model that adapts resources to channel conditions based on CQI can be written as Ri=ai.CQIi∑j=14aj.CQIj×R, where CQIi is the average CQI of cluster i. a is a priority factor that ensures fairness. This attribute ensures higher throughput while maintaining service quality for weaker UEs.

LTE-A CCa estimation model using packet bundling and carrier aggregation

Physical resources is one of the smallest entities to schedule a voice packet. If the voice packet is smaller than the physical resource element, then some portion of the physical resources may be wasted. Hence, it is possible to bundle multiple voice packets for transmission in a single physical resource element. The consumption of radio resources varies depending upon the radio channel condition. For example, in good radio condition, a packet may require fewer radio resources due to the use of a higher modulation scheme; in such a case, more than one packet can be bundled together. However, the same may not be possible in weaker radio conditions. Therefore, we can apply a packet bundling mechanism in favorable channel conditions, taking into account the channel quality. Furthermore, we can bundle the number of packets based on the radio condition. For instance, up to three or five packets can be bundled together when using 64QAM or 256QAM modulation in very good radio conditions. In average radio conditions, up to two packets can be bundled together when using 16QAM and the AMR-WB voice codec. Again, this packet bundling technique is influenced by the voice codec being used for encoding the voice packet. This part talks about analytical models that can be used to figure out how many subscribers an LTE-A cell can handle while taking into account different factors.

The number of resources required for transmitting voice packets of subscribers can be represented as Senapati & Pati (2019b),

(10) RBccreqd=(1+Ravr_re)×Nave×(1−p)×v+Nave′×p,

In general, the total number of mobile voice users can be served within the network is RBtotalRBreqd. Now, the number of mobile voice users that an LTE-A network can support in UL/DL can be expressed as:

(11) Nsupcc=RBtotal(1+Ravr_re)×Nave×(1−p)×v+Nave′×p

As per the assumption considered in “Related Works” of this article, the radio condition varies in different clusters based on the position of equipment within an LTE-A cell. Further, based on their CQI report, the modulation schemes being used also vary. Hence, the capacity of an LTE-A network within each cluster also varies. It is quite obvious that the cluster positioned near the base station may require fewer radio resources to transmit a voice packet as compared to the other clusters positioned far away from the base station. Therefore, it is possible to optimize the use of radio resources by utilizing packet bundling. Now the packet bundling factor ω can be used in the model presented in Eq. (11) to further estimate the capacity of the LTE-A network in DL. We ignore packet bundling when estimating capacity in UL because it has no impact on UL. Now, the LTE-A capacity in UL and DL using the packet bundling factor ω can be estimated as presented in Eq. (12).

(12) Nsupcc=RBtotal[(1+Ravr_re)×ω×Nave×(1−p)×v+Nave′×ω′×p]+RBtotal[(1+Ravr_re)×Nave×(1−p)×v+Nave′×p]

Nsupcc is the total number of supported users in a single cell. In general, it is the fraction of total number of resources available to carry voice packets and the required number of resources for voice packet transmission. Equation (12) is derived based on this concept. Now, the capacity of an LTE-A cell considering packet bundling and CA can be estimated as presented in Eq. (13).

(13) Nsup=n×Nsupcc

where n denotes the number of carrier components needs to be aggregated by an eNB in an LTE-A network.

Now, the number of users supported in Silver class, Platinum class, Gold class, and Diamond class can be determined through Eqs. (14) to (17).

(14) NsupSilver=RBtotal[(1+Ravr_re)×ωQPSK×Nave×(1−p)×v+Nave′×ωQPSK′×p]+[(1+Ravr_re)×Nave×(1−p)×v+Nave′×p]

(15) NsupPlatinum=RBtotal[(1+Ravr_re)×ω16QAM×Nave×(1−p)×v+Nave′×ω16QAM′×p]+[(1+Ravr_re)×Nave×(1−p)×v+Nave′×p]

(16) NsupPlatinum=RBtotal[(1+Ravr_re)×ω16QAM×Nave×(1−p)×v+Nave′×ω16QAM′×p]+[(1+Ravr_re)×Nave×(1−p)×v+Nave′×p]

(17) NsupDiamond=RBtotal[(1+Ravr_re)×ω256QAM×Nave×(1−p)×v+Nave′×ω256QAM′×p]+[(1+Ravr_re)×Nave×(1−p)×v+Nave′×p]

Now, packet bundling factor ( ω) can be defined based on the cluster (within the cluster all equipment’s generating CQI with in a same range) and their corresponding modulation and voice codec schemes used for voice transmission within that cluster. The factor ω can be determined by modulation and codec selection algorithm presented in Algorithm 2.

Algorithm 2 LTE-A modulation selection and capacity estimation algorithm.

1: UERS← UE measures RS from eNB.	
2: UECQI← SINR based CQI is calculated by UE.	
3: if ( UECQI∈ [1,3]) then /* Silver class Cluster*/	
4: MCS←QPSK	
5: Calculate the number of users supported in Silver class using Eq. (14).	
6: else if ( UECQI∈ [4,6]) then /* Platinum class Cluster */	
7: MCS←16QAM	
8: Calculate the number of users supported in platinum class using Eq. (15).	
9: else if ( UECQI∈ [7,11]) then /* Gold class Cluster */	
10: MCS←64QAM	
11: Calculate the number of users supported in Gold class using Eq. (16).	
12: else if ( UECQI∈ [12,15]) then /* Diamond class Cluster */	
13: MCS←256QAM	
14: Calculate the number of users supported in Diamond class using Eq. (17).	
15: end if	
16: Nsupcc = f1×NsupSilver + f2×NsupPlatinum + f3×NsupGold + f4×NsupDiamond	
17: Nsup=n×Nsupcc	

In the proposed algorithm, the packet bundling factor ω is assumed to be uniform across users in the same cluster. However, the bundling factor can be tuned for individual users within a cluster based on their specific channel conditions and codec requirements. The adaptive ω can be expressed as ωi=f(CQIi,SINRi,MCSi). Where CQIi is the reported CQI value of user i, SINRi is the signal-to-interference-plus-noise ratio, MCSi is the modulation and coding scheme, and f(.) is a function that assigns an optimal ω based on the user’s conditions. Mathematically, wi can be expressed as wi=wmax×CQIiCQImax. Using this model, the users with better CQI may get a higher ω and thereby improve efficiency.

LTE-A modulation selection and capacity estimation algorithm

Different modulation and coding selection algorithms are presented in the literature, and basically those are used to improve the throughput of the network. In this article, a novel modulation and voice codec selection algorithm is proposed to improve the capacity of a network in terms of accommodating voice subscribers. The algorithm is developed using MCS-CQI mapping. The procedure for modulation selection and capacity estimation using packet bundling at downlink is presented in Algorithm 2. The Algorithm 2 is executed as follows.

In Step 1, user equipment measures reference signal form eNB.

In Step 2, CQI is calculated by user equipment base on the SINR.

In Step 3 and 4, based on the CQI, QPSK is assigned to the Silver class Cluster.

In Step 5, capacity at Silver class Cluster is estimated considering all the resources are assigned to that cluster.

In Step 6 and 7, Based on the CQI, 16QAM is assigned to the Platinum class Cluster.

In Step 8, capacity at Platinum class Cluster is estimated considering all the resources are assigned to that cluster.

In Step 9 and 10, based on the CQI, 64QAM is assigned to the Gold class Cluster.

In Step 11, capacity at Gold class Cluster is estimated considering all the resources are assigned to that cluster.

In Step 12 and 13, based on the CQI, 256QAM is assigned to the Diamond class Cluster.

In Step 14, capacity at Diamond class Cluster is estimated considering all the resources are assigned to that cluster.

In Step 16, LTE-A cell capacity in terms of accommodation of subscribers is estimated considering a factor f. The factor f indicates the percentage of resources distributed in each cluster.

Finally in Step 17, the capacity is estimated using CA.

Resource distribution criteria and model parameter

In this section, numerical results for LTE-A and CCa using CA and packet bundling are presented. The results are obtained by using appropriate MCS and AMR-WB voice codecs. This section is organized as follows. First, MCS distribution criteria are presented, then model parameters and assumptions are presented, and finally, the results obtained are discussed.

Distribution criteria

When things happen in real time, link adaptation is used in LTE-A to let the eNB choose the best modulation and coding scheme based on the channel quality, which is found by the CQI and reported by the UE. According to the work reported in 3GPP (2013), the CQI index consists of 16 values, ranging from 0 to 15. The CQI index 0 is not used. When the CQI index is between 1 and 3, 4 to 6, 7 to 11, and 10 to 15, the corresponding MCS are QPSK, 16QAM, 64QAM, and 256QAM, in that order, to get the best performance. Therefore, in this work, we have considered these modulation schemes for LTE-A capacity estimation. Using 256 QAM may lead to a high bit error rate and significant retransmission overhead. This is a limitation of our study. However, in this study, we have considered a scenario and presented the quantitative results while using the modulation schemes in conjunction with different voice codecs.

Model parameters and assumptions

A single LTE-A cell is considered in this work. The cell is categorized into four clusters (i.e., Silver class, Platinum class, Gold class, and Diamond class) based on their positions from eNB as shown in Fig. 1. In the Silver class cluster, the position of UE is at the cell edge. In this case, the UE generates CQI index in between 1 to 3 based on the SINR value. In the Platinum class cluster, UEs are positioned at the middle distance between eNB and the cell edge. In this case, the UE generates CQI between 4 and 6, based on the SINR value. In the Gold class cluster, UEs are close to eNB and generate CQI in the range of 7 to 11 based on the SINR value. In the Diamond class cluster, UEs are very near to the eNB, and it generates CQI within the range of 12 to 15. An appropriate MCS is assigned by eNB to UE depending on the CQI report. The bandwidths used for capacity estimation within the cell are mentioned in Table 2. The radio conditions inside the cell determine which class is used. The Silver class uses QPSK for CQI between 1 and 3, the Platinum class uses 16QAM for CQI between 4 and 6, the Gold class uses 64QAM(5/6) modulation for CQI between 7 and 11, and the Diamond class uses 256QAM for CQI between 12 and 15. The voice codecs with bit rates used are AMR-WB 12.65, 8.85, and 6.60 kbps. The number of hybrid automatic repeat request (HARQ) re-transmissions used are 0, 1, 2, and 3 for each case scenario. To make the analysis simpler, we have considered the average re-transmission for all the clusters as equal (i.e., Ri∀iavr_re = Ravr_re) within the defined range. The average re-transmissions (i.e., Ravr_re) are used in further numerical calculations and are within the range of 0 to 3. In this work, fi is set to 0.25 as per the system parameter defined in Table 2.

Table 2 Symbols, respective values of the system parameters and detailed description.

Symbol	System parameters	Description	
Layout	1 Cell	Single LTE-A macro cell	
BWcc	1.4  3  5  10  15  20	Bandwidth in MHz	
NRB_cc	6  15  25  50  75  100	PRBs in 0.5 ms slot	
Cr	AMR-WB 12.65 kbps, 8.85 kbps, 6.60 kbps	AMR-WB voice codecs	
MCS	QPSK, 16QAM, 64QAM, 256QAM	Modulation Scheme	
Vt	20 ms	Length of the voice frame	
Ov	3 bytes	Overhead considered for voice packet	
Oh	4 bytes	Overhead considered for ROHC	
Ravr_re	0, 1, 2, 3	Average retransmission	
f	f1 = f2 = f3 = f4 = 0.25	Radio resource distribution factor	
ω	1, 0.5, 0.33, 0.17	Bundling factor for voice packet	
ω′	1, 0.33, 0.2, 0.1	Bundling factor for SID packet	
n	1, 2, 3, 4, 5	Number of carrier component	

The proposed algorithm allocates all the resources for data transmission. However, the control channel overhead may be considered, and the remaining resources may be allocated for data transmission. The actual usable resources for data transmission can be computed as RBtotal_cc=Rtotal−Rc, where RBtotalcc is the total available resource blocks for data transmission, Rc is resources allocated to control channels (Physical Downlink Control Channel (PDCCH), Physical channel HybridARQ Indicator Channel (PHICH), Physical Control Format Indicator Channel (PCFICH)), and Rtotal is the total available resource blocks.

Numerical results

This section presents the outcome of the suggested LTE-A capacity algorithm when the system parameters shown in Table 2 are used. The findings in this study show the LTE-A CCa in terms of how many voice subscribers can be supported in a single LTE-A cell, both in a simple case and when packets are bundled together. The results are obtained for each of the clusters, which is presented in Figs. 2, 3, 4, 5. Finally, the LTE-A cell capacity considering all the clusters is presented in Fig. 6.

Figure 2 LTE-A cell capacity in silver class cluster.

Figure 3 LTE-A cell capacity in platinum class cluster.

Figure 4 LTE-A cell capacity in gold class cluster.

Figure 5 LTE-A cell capacity in diamond class cluster.

Figure 6 LTE-A overall cell capacity.

LTE-A CCa in silver class cluster

The LTE-A cell capacity is estimated within the Silver class cluster using the proposed model presented in Eqs. (9) and (13) for a simplistic scenario as well as using the packet bundling approach, respectively. The results are presented in Fig. 2, which shows the capacity comparison in a simplistic scenario as well as packet bundling while using the QPSK modulation scheme w.r.t. different AMR-WB voice codecs across the bandwidths available for LTE-A. Further, the capacity is also estimated using CA for n = 2 to 5. From the results presented in Figs. 2A–2C, it is observed that, under the Silver class cluster, AMR-WB 12.65 and 8.85 give the same result, whereas the lower-order voice codec, i.e., AMR-WB 6.60 gives a significantly better result.

LTE-A cell capacity in platinum class cluster

The LTE-A CCa is estimated within the Platinum class cluster using the proposed model presented in Eqs. (9) and (13) for a simplistic scenario as well as using the packet bundling approach, respectively. The results are presented in Fig. 3, which shows the capacity comparison in a simplistic scenario as well as packet bundling while using the 16QAM modulation scheme w.r.t. different AMR-WB voice codecs across the bandwidths available for LTE-A. Further, the capacity is also estimated using CA for n = 2 to 5. From the results presented in Figs. 3A–3C it is observed that, under the Platinum class cluster, AMR-WB 12.65 and 8.85 give the same result, whereas the lower-order voice codec, i.e., AMR-WB 6.60 gives a significantly better result. Further, it is observed that the packet bundling factor gives significantly better capacity as compared to the simplistic approach. Furthermore, it is observed that the Platinum class occupancy is more than the Silver class occupancy in terms of the number of active voice subscribers.

LTE-A cell capacity in gold class cluster

The LTE-A CCa is estimated within Gold class cluster using the proposed model presented in Eqs. (9) and (13) for simplistic scenario as well as using packet bundling approach respectively. The results are presented in Fig. 4, which shows the capacity comparison in simplistic scenario as well packet bundling while using 64QAM modulation scheme with respect to different AMR-WB voice codecs across the bandwidths available for LTE-A. Further, the capacity also estimated using CA for n = 2 to 5. From the results presented in Figs. 4A–4C it is observed that, under Gold class cluster, the lower order voice codec, i.e., AMR-WB 6.60 gives significantly better result then higher order voice codecs. Further it is observed that packet bundling factor gives significantly better capacity as compared to the simplistic approach. Furthermore it is observed that, the Gold class occupancy is more than Platinum class occupancy in terms of number of active voice subscribers.

LTE-A cell capacity in diamond class cluster

Table 2 is presented for system parameters considered for this work. The LTE-A CCa is estimated within Diamond class cluster using the proposed model presented in Eqs. (9) and (13) for simplistic scenario as well as using packet bundling approach respectively. The results are presented in Fig. 5, which shows the capacity comparison in simplistic scenario as well packet bundling while using 256QAM modulation scheme w.r.t. different AMR-WB voice codecs across the bandwidths available for LTE-A. Further, the capacity also estimated using CA for n = 2 to 5. From the results presented in Figs. 5A–5C it is observed that, under Diamond class cluster, the lower order voice codec, i.e., AMR-WB 6.60 gives significantly better result then higher order voice codecs. Further it is observed that packet bundling factor gives significantly better capacity as compared to the simplistic approach. Furthermore it is observed that, the Diamond class occupancy is more than Gold class occupancy in terms of number of active voice subscribers.

LTE-A overall cell capacity

The LTE-A CCa is estimated within Diamond class cluster using the proposed algorithm presented in Algorithms 1 and 2 for simplistic scenario as well as using packet bundling approach respectively. The results are presented in Fig. 6, which shows the capacity comparison in simplistic scenario as well packet bundling. Here the overall LTE-A CCa is obtained by distributing the resources among cell clusters using the factor f. The LTE-A CCa is estimated using CA using n = 2 to 5. From the results presented in Figs. 6A–6C it is observed that, lower order voice codec gives better result then higher order voice codecs. Further it is observed that packet bundling factor gives significantly better capacity as compared to the simplistic approach. It is also observed that the radio resource distribution f can be adjusted as per the traffic condition in different clusters to optimize the capacity. Further, f can be used to analyze the data offloading, which improves the network efficiency as it is reported in Agamy & Mohamed (2021).

Conclusions and future work

LTE-A capacity in terms of voice subscribers is modeled in this article by logically dividing the LTE-A cell into four clusters, each belonging to a geographical area in which UE generates a certain level of CQI. From the results, it is observed that LTE-A capacity in each of the clusters varies depending on the radio condition of that location. The packet bundling factor ω is used in the model to optimize the radio resource usage in each cluster. The proposed algorithm assumes the radio distribution factor fi to reflect the percentage of resources allocated in a different cluster. Here the model is tested by distributing the radio resources equally in all the clusters. However, fi can be dynamically adjusted depending upon the traffic load condition of the cluster to enhance resource utilization and to reduce call blocking. The following are some of the significant observations to achieve optimal capacity. The LTE-A cell capacity using Packet bundling gives better result as compared to the model presented for simplistic scenario.

In Diamond, Gold, and Platinum class cluster, optimum capacity can be achieved by using the proposed modulation selection and capacity estimation algorithm. The bundling factor ω is found very efficient while using lower order voice codec as compared to higher order voice codec. However, ω is not having any effect in Silver class cluster.

Carrier aggregation can be used for bandwidth expansion and thereby enhancing the accommodation of active subscribers within an LTE-A cell, which is evident from the results presented in Figs. 2–6.

Loading upper bound in terms of resource utilization by the subscribers are the same. But the resource requirements for different subscribers vary with respect to the region they belong to. Hence in this study, we have tried to enhance the accommodation of subscribers within the cell based on their individual resource requirements.

Finally, it is concluded that the LTE-A capacity varies with respect to clusters depending on the radio conditions. Therefore estimating LTE-A capacity becomes a nontrivial task. The extension of this work is to include, The impact of UEs’ mobility across the clusters within an LTE-A cell needs to be investigated.

The proposed models can be extended to analyze the impact of data offloading for improving the efficiency of network.

The proposed algorithms can be further modifier suitably to incorporate QoS-aware scheduling to prioritize voice over other latency-sensitive packets.

The algorithms and models proposed in this article can be further extended to estimate the users capacity in the context of 5G network.

The base algorithm proposed in this article may be modified suitably by incorporating machine learning based approaches to predict traffic demand and dynamically adjust resource allocation in real time. This ensures optimized spectral efficiency, fair distribution, and improved QoS, making the LTE-A network more robust to variations in traffic load, user mobility, and channel conditions.

Supplemental Information

Supplemental Information 1 LTE-A Cell Capacity Estimation Data.

Each data point indicates the capacity estimated in the respective radio condition using the proposed algorithm.

Supplemental Information 2 Algorithm 1.

Supplemental Information 3 Algorithm 2.

Abbreviations

4G 4th Generation

5G 5th Generation

BW bandwidth

VoLTE Voice over Long Term Evolution

UE user equipment

GSM Global System for Mobile communications

CQI Channel Quality Indication

3GPP Third Generation Partnership Project

LTE Long Term Evolution

DL downlink

UL uplink

AMR Adaptive Multi-Rate

AMR-WB Adaptive Multi-Rate Wideband

PRB Physical Resource Block

CAC Call Admission Control

VoIP Voice over Internet Protocol

QoS Quality of Service

HARQ Hybrid automatic repeat request

eNB eNodeB

MCS Modulation and Coding Scheme

MAC Medium Access Control

AMC Adaptive Modulation and Coding

BLER block error rate

ROHC RObust Header compression

LTE-A LTE-Advanced

CA Carrier Aggregation

CC Component Carrier

CCa Cell Capacity

SINR signal-to-noise-plus-interference ratio

PCC primary component carrier

SCC secondary component carrier

QPSK Quadrature Phase Shift Keying

16QAM 16 Quadrature Amplitude Modulation

64QAM 64 Quadrature Amplitude Modulation

256QAM 256 Quadrature Amplitude Modulation

MIMO Multiple Input Multiple Output

RF radio frequency

NR New Radio

Additional Information and Declarations

Competing Interests

The authors declare that they have no competing interests.

Author Contributions

Rajiv Senapati conceived and designed the experiments, performed the experiments, analyzed the data, performed the computation work, prepared figures and/or tables, authored or reviewed drafts of the article, and approved the final draft.

Data Availability

The following information was supplied regarding data availability:

The data and code are available in the Supplemental File.

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
