# Peer review of "Modeling LTE-advanced cell capacity estimation using packet bundling and carrier aggregation"

_PeerJ Computer Science, doi:10.7717/peerj-cs.2868_

## Round 0.1 · original submission · Major Revisions

Please clarify your contribution in the paper. Rewrite the abstract to highlight the contribution. You need to follow all the comments written by the reviewers

Reviewer 1 ·

Basic reporting

Review the grammar and overall English writing throughout the paper for clarity, coherence, and correctness.

Experimental design

1. The main contribution of the author is to optimize the radio resource utilization by dividing the LTE-A cell logically into clusters such as Silver class, Platinum class, Gold class, and Diamond class. These clusters are formed based on the reported CQI by the UE to eNB.
In the manuscript, the author has failed to explain about the clustering of Silver class, Platinum class, Gold class, and Diamond class depending upon the CQI. There is no valid calculation for determining CQI values for the scenario which you have mentioned.
In the revised manuscript the author should concentrate on calculating the CQI values (Not only based on SINR – Include all parameters) for the scenario prescribed and how the clusters are formed. Include analytical expressions for estimating CQI.
2. The algorithm assumes fixed percentages of resources allocated to each cluster, which may not reflect real-time traffic demand and channel conditions. This rigid allocation may result in underutilization or overloading of resources in certain clusters.
Dynamic resource allocation mechanism should be used in real time based on traffic demand, channel conditions, and user distribution. If this is not done the author should justify how the proposed algorithm is efficient.
3. The algorithm assigns MCS (e.g., QPSK, 16QAM, 64QAM, 256QAM) based on static CQI ranges without considering advanced link adaptation techniques, such as Hybrid ARQ (HARQ) or feedback-based adjustments.
more adaptive MCS assignment that incorporates real-time CQI feedback and considers retransmissions through HARQ to optimize throughput and reliability.
4. The packet bundling factor (ω) is assumed to be uniform across users in the same cluster, which does not account for user-specific variations in channel conditions or codec efficiency.
Refine the bundling mechanism by allowing ω to vary for individual users within a cluster based on their specific channel conditions and codec requirements. This can further enhance resource efficiency.
5. The algorithm assumes all resources are available for data transmission, neglecting the impact of control channel overhead (e.g., PDCCH, PHICH) on overall capacity. Incorporate control channel overhead in the capacity calculations to ensure a more accurate estimation of available resources for data transmission.
6. There is no mechanism to prioritize critical packets (e.g., voice packets over data packets) in resource allocation or bundling. Try to add Quality of Service (QoS)-aware scheduling to prioritize voice and other latency-sensitive packets.

Validity of the findings

The results and discussion is very weak. The analysis should be expanded by including parameters such as spectral efficiency, throughput, latency, packet loss rate, energy efficiency, and QoS metrics to provide a comprehensive evaluation of the algorithm's performance.

Additional comments

Enhance the algorithm by Implementing a resource scheduler that adapts to dynamic channel conditions, user density, and interference. Use predictive models to optimize MCS selection, resource allocation, and CA configurations.

Reviewer 2 ·

Basic reporting

I have the following comments of the manuscript titled "Modeling LTE-advanced cell capacity estimation using packet bundling and carrier aggregation".

1. What is the claim of Author to call as LTE-Advanced as the topic discusses correspond to LTE syste.
2. How packet bundling is different from carrier aggregation.
3. Validate Eq. (1) as the modulation order is less for LTE systems.
4. In main contribution of Introduction section, the novelty of the authors is not explicit as the claim is the available system.
5. What is the 3GPP release the author refer to for packet bundling?
6. The variables of the equations are not clearly defined. For instance, the Eqs.(2) to Eq. (6) is not clear and not proper citation is available.
7. Section 4.1.1 has to be completely revised as it does not clearly provide insight about the proposed algorithm.

Experimental design

1. The details of experiments and the tool used is not clear.
2. The system model is not provided
3. The constraints of the LTE-A is not provided.

Validity of the findings

1. How packet bundling is different from carrier aggregation. The details need to be included.
2. In main contribution of Introduction section, the novelty of the authors is not explicit as the claim is the available system.

---

## Round 0.2 · accepted · Accept

Congratulations to the authors for addressing all the issues raised by reviewers. The paper is ready for publication.

Reviewer 2 ·

Basic reporting

The authors have responded to my earlier comments

Experimental design

The design is appropriate

Validity of the findings

The findings are good

Additional comments

None